# An Innovative Decision Support System to Improve the Energy Efficiency of Buildings in Urban Areas

**Małgorzata Sztubecka** [1] , **Marta Skiba** [2,*] , **Maria Mrówczyńska** [2] and **Anna Bazan-Krzywoszańska** [2]

1. Faculty of Civil and Environmental Engineering and Architecture, UTP University of Science and Technology, ul. prof. S. Kaliskiego 7, 85-796 Bydgoszcz, Poland; sztubecka@utp.edu.pl
2. Faculty of Civil Engineering, Architecture and Environmental Engineering, University of Zielona Góra, ul. prof. Z. Szafrana 1, 65-516 Zielona Góra, Poland; M.Mrowczynska@ib.uz.zgora.pl (M.M.); A.Bazan@aiu.uz.zgora.pl (A.B.-K.)
* Correspondence: M.Skiba@aiu.uz.zgora.pl

**Abstract:** Improving in the energy efficiency of urban buildings, and maximizing the savings and the resulting benefits require information support from city decision-makers, planners, and designers. The selection of the appropriate analytical methods will allow them to make optimal design and location decisions. Therefore, the research problem of this article is the development of an innovative decision support system using multi-criteria analysis and Geographic Information Systems (decision support system + Geographic Information Systems = DGIS) for planning urban development. The proposed decision support system provides information to energy consumers about the location of energy efficiency improvement potential. This potential has been identified as the possibility of introducing low-energy buildings and the use of renewable energy sources. DGIS was tested in different construction areas (categories: A, B, C, D), Zielona Góra quarters. The results showed which area among the 53 quarters with a separate dominant building category was the most favorable for increasing energy efficiency, and where energy efficiency could be improved by investing in renewable energy sources, taking into account the decision-maker. The proposed DGIS system can be used by local decision-makers, allowing better action to adapt cities to climate change and to protect the environment. This approach is part of new data processing strategies to build the most favorable energy scenarios in urban areas.

**Keywords:** energy efficiency of buildings; energy potential; renewable energy sources; Geographic Information System; multi-criteria analysis; smart city; urban analysis

---

## 1. Introduction

Progressing urbanization and climate change are interacting processes that require comprehensive actions to minimize their negative effects. Particularly significant are the challenges related to the intensity of threats of the progressive dispersion of housing construction along its communication routes (urban sprawl) and the increasing socio-environmental fragmentation of urban areas [1–3]. These threats often result from a lack of knowledge and in-depth analysis at the local level, starting from a single building, through housing estates, and ending in the areas of basic territorial units; i.e., municipalities. Therefore, this article analyzes the improvement of energy efficiency in urban areas, starting from individual households in buildings (including multi-family buildings) located in quarters characterized by similarity of structure and intensity of development.

Urban areas, which gather more than half of the world's population [4], have a steadily increasing share in global energy consumption, and all indications are that this trend will not change. At the

same time, cities are centers of innovation, engines of development and economic growth, and provide the necessary services to their residents. Energy consumption in urban areas results directly from their function, the way buildings are used, and the increasing time we spend in them. Also, It results from the growing demand for services (including construction), and the need to ensure and maintain thermal comfort in buildings.

To reduce energy consumption in cities (reduce environmental pollution) at a supranational level, a new development path has been proposed for cities based on their adaptation to climate change: the report, Urban Adaptation to Climate Change in Europe in 2016. Transforming Cities in a Changing Climate [5,6] is an overview of the actions that can be taken to adapt cities in Europe with the future challenges faced by cities due to the effects of climate change. The proposals relate to changes in spatial planning at the local level, and in the buildings, designs to make cities more economical, less energy-consuming, safe, and more attractive to their residents. The growing popularity of design methods that create buildings with low energy consumption (low energy, passive, zero, and plus energy buildings) [7–9], as well as new technologies using renewable energy (not increasing greenhouse gas emissions), such as photovoltaic panels, solar collectors, wind turbines, and heat pumps significantly improve energy efficiency. The work of a research team from Kuwait [10] showed the possibility of transforming a public utility building into a zero energy building by introducing as a source of energy, photovoltaic systems mounted on a roof in three energy scenarios (energy savings of over 1400 MWh per year, while reducing $CO_2$ emissions by over 700 tons per year). Energy scenarios have also been used in a study of residential buildings. By indicating the possibility of using solar energy to transform residential homes into zero energy, profitability in terms of emissions can be achieved in less than two years [11]. Baljit et al., when analyzing the efficiency of photovoltaic syfigurestems for roofs and walls of buildings also indicated the need to integrate the technology of erecting buildings with these systems. At the same time, they postulate that economic aspects and climate conditions should be considered as a priority when designing construction objects [12].

The wind energy potential is currently used mainly in coastal and submontane areas as well as in open areas with favorable conditions [13]. It is worth emphasizing, however, that technologies are being developed to obtain wind energy in built-up areas. Rezaeiha et al. indicated in research the possibility of using wind turbines with vertical axes in the urban environment, adapted to changing wind directions (common in built-up environments) [14]. Systems for properly designed heat pumps are another way to generate energy in urban areas. They allow heat transfer to the building from the surrounding air (also moist) and energy gain during the day and at night [15]. Combining early, at the design stage, conventional energy sources and energy obtained from renewable sources immunizes systems against climate change. Bagheri et al. for that purpose proposed the creation of urban hybrid renewable energy systems that through optimal planning, will allow the use of energy on a neighborhood scale, taking into account the real-time of the electric load of the city [16]. An interesting solution proposed in Polish work [17] is the creation of energy cooperatives based on local communities that produce energy using renewable energy sources and use it for their own needs. Such local policy increases social and environmental awareness and can improve the quality of residents' lives.

*Background*

Energy efficiency is constantly increasing in newly designed buildings and is improving in existing ones, where, along with changes in the living standard, the following are created: thermomodernization, changes in the heating method and preparation of hot water, and replacement of equipment and lighting. Programs for the above improvements are most often associated with benefits from support (also financial) programs. There are many possible changes and improvements, and each brings different benefits, so what is the possibility of choosing the optimal variant? It should also be remembered that the programs have different recipients—building owners, users using RES (renewable energy source), electricity producers, users of some technology or obtaining a specific consumption rate, prosumers, organizations related to environmental protection, etc. Also, not all changes can be made

in all locations (some depend on climate, wind strength, thermal water temperature, distance of the power network and others that can be effectively identified using geoinformation tools). There are also significant differences between types of buildings and energy consumption and production profiles and the months in which they reach their maximum values. The greatest impetus for energy efficiency measures is undoubtedly economics. The question, therefore, remains whether policymakers have sufficient knowledge and information about the economic benefits of using energy efficiency improvements [18].

Energy demand monitoring in cities is the task of municipalities, which while implementing the spatial policy (at the lowest level) of the country, should strive to limit the use of energy and greenhouse gas emissions [19]. The local government decision-makers need effective tools to achieve this goal, which will support making decisions regarding both spatial planning and low-energy building design, as well as introduce renewable energy sources into urban areas. The article aims to develop an innovative system supporting the decision-making process in local government institutions, which uses multi-criteria analysis and GIS (geographic information system) (DGIS – decision support system for GIS). Multi-criteria analysis and mathematical methods of its implementation in practical issues related to construction and improvement of energy efficiency are currently a very dynamically developing field of knowledge. To assess the energy efficiency of low and zero energy buildings, Hu, in the article [20], proposed the use of life cycle and multi-criteria analysis. In this way, he identified additional benefits related to a building's energy efficiency, such as improving human health and protecting the environment. Using the method he developed to assess the relationship between building and energy consumption, environmental impact (smog, global warming, ozone reduction), water (acidification, eutrophication), and human health, he found alternative construction and material solutions with low environmental impact have a more beneficial effect on human health but do not always correlate with energy savings over the building's life cycle. Launay et al. proposed the use of multi-criteria analysis to optimize interseason solar energy storage to meet the energy needs of residential buildings. The authors analyzed storage systems in terms of meeting the demand for heating and hot water [21]. Kozik et al. present the method of selecting the most advantageous material solutions supported by the multi-criteria analysis method. In their work, they described a method for selecting the best thermal insulation and thermo-modernization of buildings for a potential investor [22]. They also wrote about the possibility of using multi-criteria analysis for the energy performance of a building or group of buildings in an article by Ziembicki et al. In addition to energy performance, the authors recommend an energy source concerning $CO_2$ emissions and predetermined construction investment costs [23]. They conducted their own research based on data on the energy efficiency of various types of buildings located in Polish cities. Ouria [24] presents research on the use of solar energy in cities and buildings, taking into account many geographical and climatic factors, aimed at estimating the potential. The author analyzed the decisive factors in the assessment of solar energy: geographical parameters, climatic factors, types of radiation, analysis of building geometry, and building orientation. Research results indicated the possibility of increasing the solar energy potential in the city with to the use of appropriate orientation and geometry of individual buildings.

Supplementing multi-criteria analysis by GIS tools allows for spatial analyzes, enables quick identification of changes and determining the impact extent of decisions related to space. Multi-criteria analysis techniques based on GIS are used, among other things, to determine the optimal building location, endangered or protected area locations, or areas designated for various investments.

The article [25] is an example of a combination of decision methods and the GIS system. It presents the method of choosing the location of the investment (municipal waste incineration plant) in terms of the principles of sustainable development. Feyzi et al. analyzed environmental, economic, and socio-cultural criteria and applied, using a geographical information system, the final weights to each of the adopted criteria. The results obtained indicate the possibility of using the proposed solution by decision-makers who want to make decisions by the principles of sustainable development. Similar results were obtained in their research by Arabameri et al. They used multi-criteria decision

analysis based on GIS for developing the methodology used by decision-makers and managers to plan preventive measures and reduce damage caused by erosion [26]. An interesting combination of the presented approach was proposed by the Erbaş et al. hybrid method supporting the decision making process regarding the location and construction of vehicle charging stations [27]. The methodology of making decisions regarding the determination of optimal locations for wind farms was also presented in [28], in which three location scenarios were analyzed—each contained a different set of restrictions. The construction of scenarios supporting GIS with multi-criteria analysis has allowed the creation of decision models that can be easily implemented in other areas and updated if there is a change in the provisions related to spatial planning and location of such investments. Optimal development of space was also analyzed by Özceylan el al. Taking into account 13 geographical criteria in their research, they proposed the best location of a freight village among 20 alternative solutions [29]. The problem of urban development and the vulnerability of urban structures to threats is more widely presented by Ghajari et al. [thirty]. The authors proposed to examine the physical susceptibility of buildings to external factors (14 vulnerability criteria were defined, and a standard environment for each of them was generated in the GIS environment) and pointed to the need to reduce the risk of threats to buildings. It seems that it is advisable to help with choosing the variant, scenario, and selection criteria for individual decisions dependent on local authorities because they should reduce the vulnerability of urban areas to physical hazards and mandate the obligation to conduct sustainable spatial policy and urban development [30].

The innovation of the proposed system is to provide decision-makers with the option of using multi-criteria analysis supported by GIS to improve the energy efficiency of buildings in urban areas. The use of GIS methods allows the visualization and analysis of many alternative development scenarios and optimal selections, which positively affects the decision-making process. Although, as mentioned above, there are already studies that use multi-criteria analysis methods based on GIS data, there is still a gap in the use of the proposed methods to determine the energy efficiency potential of urban areas taking into account technological, economic, urban, and social criteria. The proposed decision support system DGIS, was tested in select quarters of Zielona Góra, a medium-sized city, located in Poland (Europe).

The energy consumption visualization of urban buildings, in the private and public sectors, can be a powerful tool to increase behavior related to economic optimization but also environmental protection. Visualizing trends, information, and the dynamics of changes can enrich recipients with new knowledge to avoid improper workload and resources; e.g., in the selection of heating systems, investing in energy efficiency measures and renewable energy sources, and in new development projects [18]. The article presents a decision support system using multi-criteria analysis and GIS, which allows mapping the results of analyzes in the city space. The article proposes innovative control of the decision-making process (depending on the level of decision-makers and appropriate transmission of media to them—the GIS tool) in a way that allows achieving the highest possible energy savings. GIS technology support guarantees quick visualization of results in a real environment, allowing for analysis of variants; i.e., a practical guide for decision-makers.

The structure of the article is as follows: Section 2 describes the materials and methods used. In that section, in addition to the assessment of the decision-making and how to make decisions about ways to improve energy efficiency in urban areas, it presents the possibility of implementing the DGIS. Part 3 presents the results and discussion, which are the result of the case study analysis of buildings in the city of Zielona Góra. The conclusions are contained in Section 4.

## 2. Materials and Methods

### 2.1. Study Area

The city of Zielona Góra is located in the west of Poland, historically associated with Western Europe, which is reflected in the planning and urban development; in particular, in the quarters located in the old/historic city and downtown. The city covers an area of 278 km$^2$, although for this research an area of 58 km$^2$ was adopted, divided into 53 quarters (see in Section 3. Results and Discussion). Zielona Góra is a medium-sized city, one of 38 Polish cities where the population exceeds 100,000. According to the latest available data (as of December 2018) the city's population is 140,297 inhabitants and the population density is 504 people/km$^2$ [31] (Figure 1).

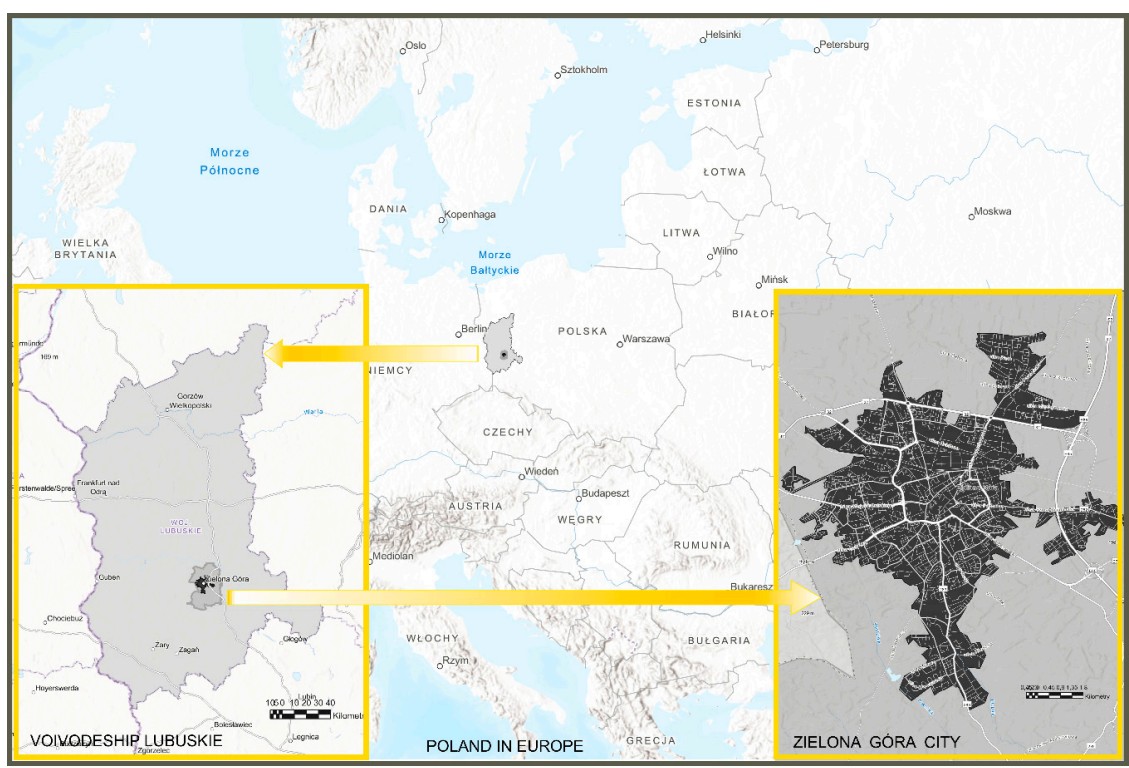

**Figure 1.** Location and specification of Zielona Góra in Poland.

The structure of Zielona Góra's development is varied. In the city center and downtown, compact, frontage buildings with traditional technology prevail, where coal burning is still the predominant way of heating residential and residential and commercial buildings. Due to the age of buildings, construction technology, and social conditions, this region is characterized by high energy consumption. The center and downtown are surrounded by housing estates built in the 1950s with traditional or industrialized technology. The majority of heating in this area is the use of heat from the district heating network (CHP—Combined Heat and Power, heat from cogeneration). Single-family, semi-detached, and terraced buildings built (from the 1980s) in traditional technology are located in the suburbs and are heated using gas, electricity, and coal [19].

### 2.2. Methodology

There are many methods of multi-criteria analysis. The most commonly used are mathematical methods, but there are also geometric methods, methods based on graph theory, taxonometric methods, and methods based on artificial intelligence; in particular, fuzzy systems and neural-fuzzy networks [32,33]. The method chosen depends on the decision problem being solved, and when

choosing a method, factors such as ease of data collection, processing and verification of results, and subjectivity of the solution obtained should be taken into account [32]. The latter factor is particularly important in making decisions on issues related to spatial planning, architecture, and construction because in these areas we use the knowledge of experts. For this reason, the obtained result may be burdened with errors.

According to the algorithm shown in Figure 2, using mathematical methods, can support the decision-making process and make the most optimal decisions in the following steps [34,35]:

- Step 1—defining the analysis goal. This is identifying the energy efficiency potential in urban areas, building energy scenarios depending on the decision-maker, and analyzing the acceptable solution set $M$:

$$M = \{M_i : i = 1, 2, \ldots, m\}; \tag{1}$$

- Step 2—defining of the initial set of criteria and analysis taking into account the representativeness of the criteria, interrelationships between the criteria, and the level of detail of the description of the subject of assessment; construction of the final set of criteria $C$ with the number $n$:

$$C = \{C_j : j = 1, 2, \ldots, n\}; \tag{2}$$

- Step 3—setting the criteria weights with the participation of the decision-maker and experts: because the criteria have unequal validity, hierarchical factors (weights) should be entered in the analysis process:

$$W = \{W_j : j = 1, 2, \ldots, n\}; \tag{3}$$

- Step 4—determination of numerical measures of individual analysis variants:

$$Z_{ij} = \{z_{ij} : i = 1, 2, \ldots, m; j = 1, 2, \ldots, n\}; \tag{4}$$

based on the determined criteria measures, a data **Z** matrix is obtained

$$\mathbf{Z} = \begin{bmatrix} z_{11} & \cdots & z_{1j} & \cdots & z_{1n} \\ \vdots & \cdots & \vdots & \cdots & \vdots \\ z_{i1} & \cdots & z_{ij} & \cdots & z_{in} \\ \vdots & \cdots & \vdots & \cdots & \vdots \\ z_{m1} & \cdots & z_{mj} & \cdots & z_{mn} \end{bmatrix}. \tag{5}$$

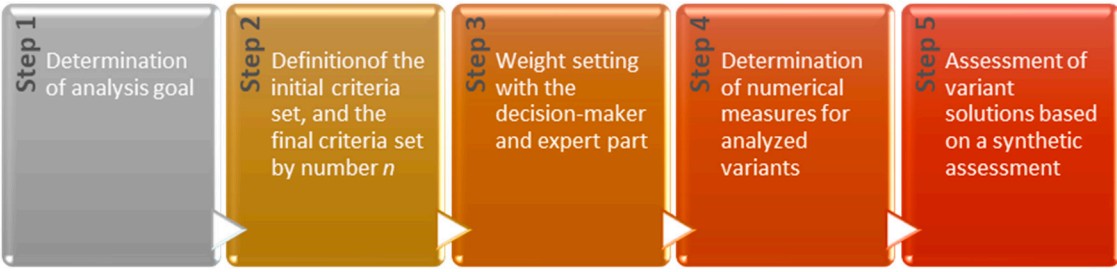

**Figure 2.** Decision support algorithm by multi-criteria analysis.

The **Z** matrix rows present partial measures of individual variants. The columns contain partial measures of all variants according to a specific partial criterion. In mathematical methods, an important element is the coding of measures adopted for analysis, aimed at replacing the appointed value of a

partial measure with an unnamed (numerical) value from a specific range. In the work, the determination of encoded partial measures of individual variants was carried out using standardization according to the following dependencies [36]:

$$z_{ij}^* = \frac{z_{ij} - \bar{z}_j}{\sigma_j} \text{ (maximizing normalization)}, \tag{6}$$

$$z_{ij}^* = (-1)\frac{z_{ij} - \bar{z}_j}{\sigma_j} \text{ (minimizing normalization)}, \tag{7}$$

where: $z_{ij}^*$—the encoded value of the partial measure $z_{ij}$, shown in the matrix (5)

$$\bar{z}_j = \frac{\sum_{i=1}^m z_{ij}}{m}, \tag{8}$$

where: $\bar{z}_j$—average measure of analyzed variants according to individual criteria

$$\sigma_j = \sqrt{\frac{\sum_{i=1}^m (z_{ij} - \bar{z}_j)^2}{m}}, \tag{9}$$

where: $\sigma_j$—standard deviation with m number of variants.

- Step 5—assessment of variant solutions based on a synthetic assessment. The result is finding the most advantageous variant in terms of the adopted criteria—the ability to make appropriate decisions. To do this, the value of each variant should be calculated using one of the synthetic formulas, which in the case of research is the adjusted summation index, defined as:

$$S_i = \sum_{j=1}^n (z_{ij} w_j), \tag{10}$$

where: *S*—value of individual variants, *z**—measure of each of the variants, and *w*—criterion weight, which is determined by the decision-maker individually or after obtaining an expert opinion.

The use of multi-criteria analysis methods allows the assessment of proposed variants of a given solution, but the assessment itself is largely dependent on the adopted criteria. Using multi-criteria analysis to optimize spatial issues, one usually encounters problems related to a large number of variants that can be used, as well as those difficult to unify and conflicting criteria that represent the different interests of social groups. Variants of spatial solutions are assessed by many people who have a different hierarchy of importance of adopted criteria. Therefore, the introduction of GIS technology to optimize spatial issues can be helpful not only as a graphic tool but also as a tool for collecting related information (also non-graphical), storing, processing, and presenting to a wide range of recipients in integrated information systems [37,38]. GIS used to prepare input data allows you to determine the form of land development based on data from aerial photographs and aerial laser scanning. In turn, visualization of modeling results allows supporting decisions regarding issues related to the location of investment or integration of activities with space to which these activities may relate. Considering the above, it should be assumed that all activities related to spatial analysis are associated with their processing using geoinformation software, which is used to obtain new spatial information. Adopting such reasoning, GIS is treated as a data processing tool aimed at supporting decision-making processes [39]. Also, data stored and made available in GIS systems are used for the presentation of BIM (building information modeling) models of individual buildings, which facilitates the analysis, provided that standardized cartographic methods of data presentation are used [40,41].

In summary, GIS offers the opportunity to analyze your data and make informed decisions based on that analysis. Combining GIS capabilities with a rich set of multi-criteria analysis methods and

procedures allow one to design, evaluate, and select alternative variants. The process of transforming and combining geographical data, supported by expert analysis, taking into account the decision-maker preferences, is one of the basic tasks of the GIS system and multi-criteria assessments [42].

The proceedings within the DGIS system can be presented in the form of phases combining computational works according to the algorithm of multi-criteria analysis and works related to the use of GIS technology:

- Phase 1—clarifying the purpose of the analysis—determining the energy potential in individual decision variants and defining the initial set of criteria, and, as a consequence, determining the final criteria set affecting energy consumption in buildings. At the same time, preparation of GIS layers using ArcGIS 10.7—vectorization of reference units and determination of the prevailing category of buildings.
- Phase 2—determining the weighting of criteria with the participation of decision-makers and experts, determining the numerical measures of individual variants subject to analysis, and coding partial measures of individual variants using standardization. Calculations were carried out using MATLAB$^{TM}$ software. Because in this study the criteria and weightings of the criteria were developed based on the opinions of decision-makers and experts, preliminarily prepared reports were forwarded to experts for assessment, improvement, and possible supplementation. All experts had knowledge and experience in the research topic.
- Phase3—analysis of the results obtained using GIS tools, which allowed verification of the results achieved depending on the decision variant, expert opinions, and the predominant category of objects located in reference units.

A detailed scheme of proceedings according to the method proposed in the article combining multi-criteria analysis and GIS technology is presented in Figure 3.

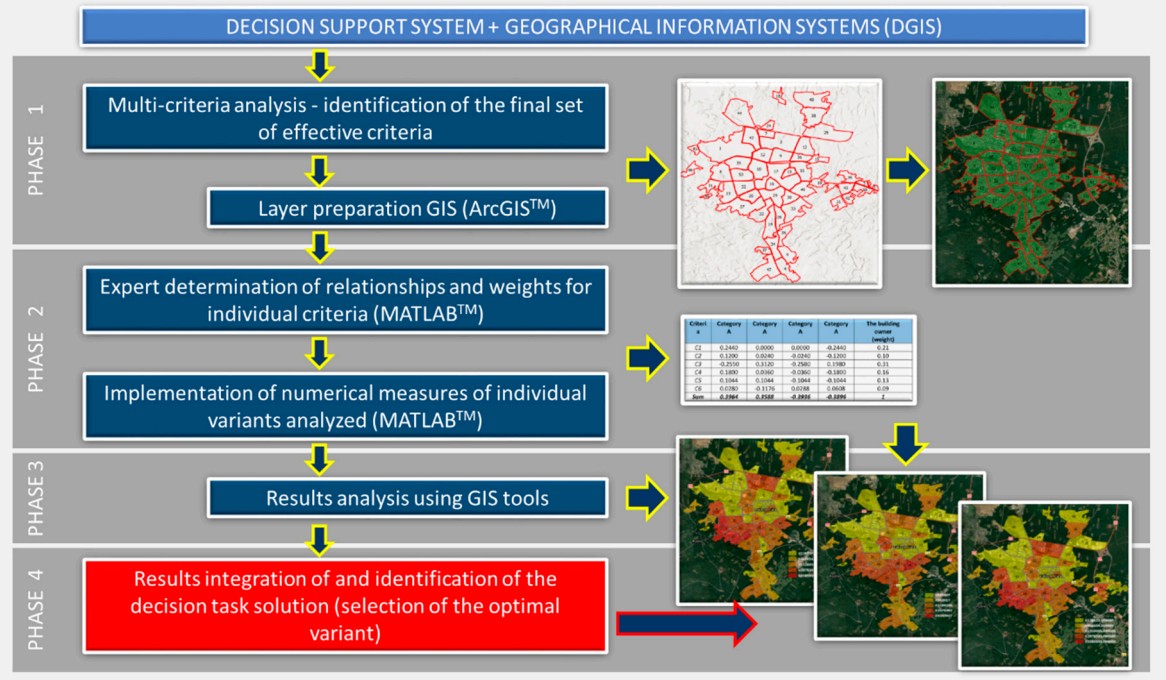

**Figure 3.** Research levels.

Undertaken research using the GIS system required the use of ArcGIS 10.7 software, which allows for conducting spatial analyses, building models and cartographic visualization of obtained test results. The work uses the PL-ETRF89 geodetic reference systems, which are the mathematical and physical

implementation of the European earth reference system ETRS89. This reference system and coordinate system used are legally defined in Poland in the 342 Regulation of the Council of Ministers [43].

## 3. Results and Discussion

Using the innovative DGIS system designed to determine the energy efficiency potential in urban areas, a case study of the city of Zielona Góra was carried out. Scenarios were built and the most favorable variants of the proposed solutions were analyzed (Table 1). When considering possible solutions, the criteria (presented in Figure 4) were adopted for analysis, which were used to assess individual variants. Before the analysis, the buildings were divided into group categories (Figure 5). The analysis of energy efficiency potential for individual buildings (quarters) was carried out for five variants, which were created depending on the decision maker's preferences (Figure 6).

**Table 1.** Input data for building scenarios in the DGIS.

| | Criteria for Buildings Energy Consumption | Building Groups Categories | Decisions Variants |
|---|---|---|---|
| **Input Data** | construction years (in quarters) | category A—multi-family buildings made in prefabricated and traditional technology | Option 1—equal validity of all criteria |
| | overwhelming feature | category B—single-family buildings made in traditional technology | Option 2—the decision-maker is the owner of the building or apartment |
| | prevailing state of ownership | category C—buildings with a service function made in traditional technology | Option 3—the local government is the decision-maker |
| | prevailing construction technology | category D—large-area industrial and service buildings constructed in prefabricated technology | Option 4—the energy company is the decision-maker |
| | predominant source of heat | | Option 5—decision-makers are representatives of organizations related to environmental protection |
| | possibility of using energy from RES | | |
| | cost of bringing 1 m$^2$ of the facility to EU requirements for energy efficiency | | |

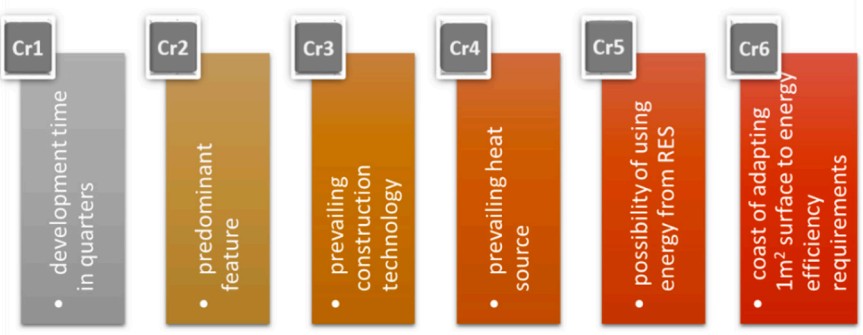

**Figure 4.** Criteria affecting energy consumption in buildings.

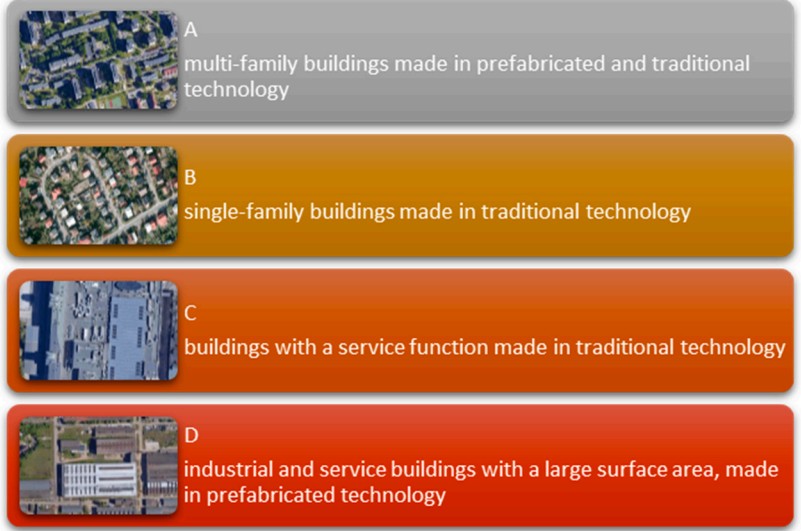

**Figure 5.** Building categories (A,B,C,D—see in Table 1) adapted for testing.

**Figure 6.** Decision variants depending on the decision-maker.

These preferences were taken into account by adjusting the weight values of individual criteria following the results obtained based on expert analyzes. Numerical measures for individual variants have been calculated using standardization according to Equations (6) and (7). Standardization was carried out for individual variants, taking into account the individual criteria for which the input values were determined as follows:

- Construction time in quarters (Cr1)— five periods picked out that represented buildings in the quarters: until 1965, 1966–1985, 1986–1992, 1993–2008, and from 2009. The time intervals were determined based on the dominant period of ongoing construction investments.
- Overwhelming function (Cr2)—four overwhelming functions have been distinguished: residential, residential-service, service, and industrial.
- Predominant manufacturing technology (Cr3)—two dominant ways of constructing objects (technology): traditional and prefabricated.
- The predominant source of heat (Cr4)—three types of urban areas have been distinguished: with a predominance of buildings heated with a solid fuel boiler, with a predominance of buildings supplied with heat by CHP, with a predominance of buildings heated with cold gas boilers.
- The possibility of using energy from renewable energy sources (Cr5)—three levels showing the willingness of decision-makers to invest in renewable energy sources and taking into account the technical possibilities of obtaining energy from renewable energy sources: low, medium, high.

- Cost of bringing 1 m² of the facility to EU requirements for energy efficiency (Cr6)—the cost has been estimated for individual categories of buildings (A, B, C, D) based on the data contained in the study [44].

Table 2 summarizes the partial measures of Variant 1, taking into account the partial criteria broken down into individual buildings categories. For the other options, summation indicators corrected in Table 3 have been compiled; calculations have been made using MATLAB™ software, according to (10). Summation indicators indicate the possibility of increasing (or decreasing) the energy potential in individual quarters due to the predominant category of buildings and the decision-maker. A weighted adder value greater than 0 indicates the possibility of achieving energy savings in a given quarter, while an adder less than 0 indicates difficulties in achieving potential energy savings. This may be due to a decision-maker not interested in making changes, or adverse technical conditions. The value of the energy potential is directly proportional to the value of the summation index obtained for individual variants.

**Table 2.** List of variants measures of the partial criteria—Variant 1.

| Criterion | Buildings Category | | | |
|---|---|---|---|---|
| | **A** | **B** | **C** | **D** |
| creation time | 0.1743 | 0.000 | 0.0000 | −0.1743 |
| function | 0.1714 | 0.0343 | −0.0343 | −0.1714 |
| ownership status | −0.1214 | 0.1486 | −0.1229 | 0.0943 |
| construction technology | 0.1714 | 0.0343 | −0.0343 | −0.1714 |
| heat source | 0.1243 | 0.1243 | −0.1243 | −0.1243 |
| use of energy from RES | 0.0500 | −0.2100 | 0.0514 | 0.1086 |
| coast of adapting 1 m² surface to UE requirements | −0.0714 | 0.2143 | −0.0714 | -0.0714 |
| **Total Adjusted Indicator** | **0.4986** | **0.3457** | **−0.3357** | **−0.5100** |

**Table 3.** The overall list of adjusted indicators—Variants 1–5.

| Total Adjusted Indicator | Buildings Category | | | |
|---|---|---|---|---|
| | **A** | **B** | **C** | **D** |
| Variant 1 | 0.4986 | 0.3457 | −0.3357 | −0.5100 |
| Variant 2 | 0.3964 | 0.3588 | −0.3936 | −0.3896 |
| Variant 3 | 0.6355 | 0.5015 | −0.3890 | −0.7490 |
| Variant 4 | 0.1651 | 0.5266 | −0.3618 | −0.3304 |
| Variant 5 | 0.3926 | −0.3021 | −0.0875 | −0.0033 |

To determine the energy potential in individual quarters, taking into account the predominant category of buildings and determinants of the decision-making situation, data obtained as a result of the energy audit for the city of Zielona Góra [44] and information contained in the study [45] were used. Based on the data contained in [44], it was estimated that the average energy potential per 1 m² is 25PLN (5.68€); in the estimation, the main element affecting the final value of the potential was the method of heating, which generates the largest cost, but at the same time, its modification can contribute to improving the energy efficiency of buildings. Calculation results, carried out for all quarters, taking into account the usable floor space in m², the predominant category of buildings (A–D), and the average energy potential m², are summarized in Table 4 and graphically presented in Figures 7–11. The highest energy potential is possible to obtain in Option 3 when the decision-maker is the local government. Such a result confirms the need to direct the financial stream aimed at improving energy efficiency to local governments. Equally, this approach gives the possibility of the most effective use of financial resources, also due to the desire to invest in renewable energy. Comparable results in the form of energy potential were obtained for Option 2 (decision-maker—owner) and Option 4

(decision-maker—energy company), which indicates that both building owners and energy producers are aware of the need to introduce changes that improve the energy efficiency of buildings. Owners, thus, strive for operational savings, and enterprises focus on the development of new technologies because they perceive the problem of market stability associated with price volatility and a steady supply of fossil fuels. The lowest energy potential is obtained if the decision-makers are people closely related to environmental protection. This is usually the result of their narrow views in a narrow field focused on reducing $CO_2$ emissions while not interested in improving the quality of residents' life.

**Table 4.** Energy potential (PLN -The Polish zloty)/(€ - Euro)—Variants 1–5.

| The Sum of m² Usable Floor Area | Energy Potential [PLN]/[€] | | | | |
|---|---|---|---|---|---|
| | Variant 1 | Variant 2 | Variant 3 | Variant 4 | Variant 5 |
| 4,233,104 | 31,909,323/ 7,252,119 | 26,996,982/ 6,135,678 | 42,587,563/ 9,678,991 | 21,119,910/ 4,799,979 | 9,621,691/ 2,186,748 |

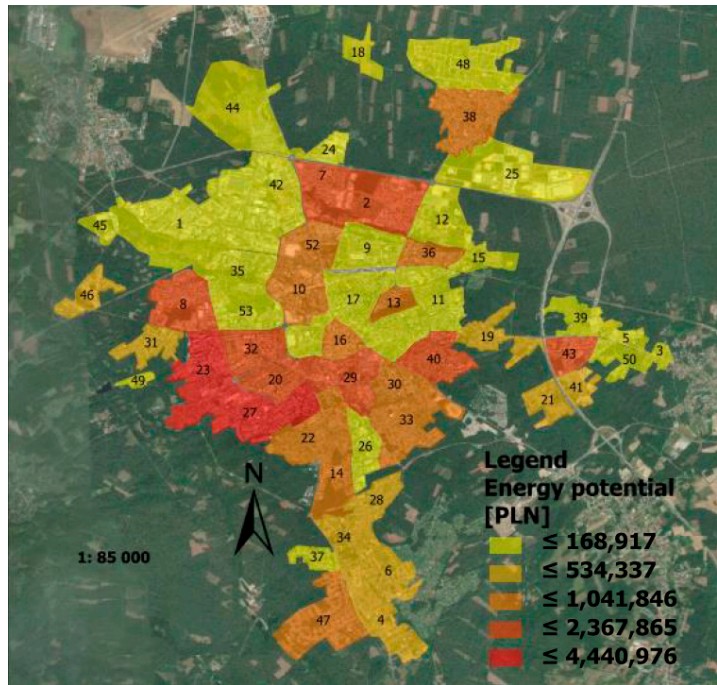

**Figure 7.** Energy potential for the Variant 1.

The proposed approach allowed the estimation of energy-saving potential in Zielona Góra, depending on the adopted criteria, including the possibility and willingness to invest in renewable energy by various groups of decision-makers. The construction of the scenarios confirmed that one can try to predict future energy usage by developing different probable but contradictory perspectives. With GIS tools, differences in the costs of their implementation can be shown in an easy (illustrative) and accessible way.

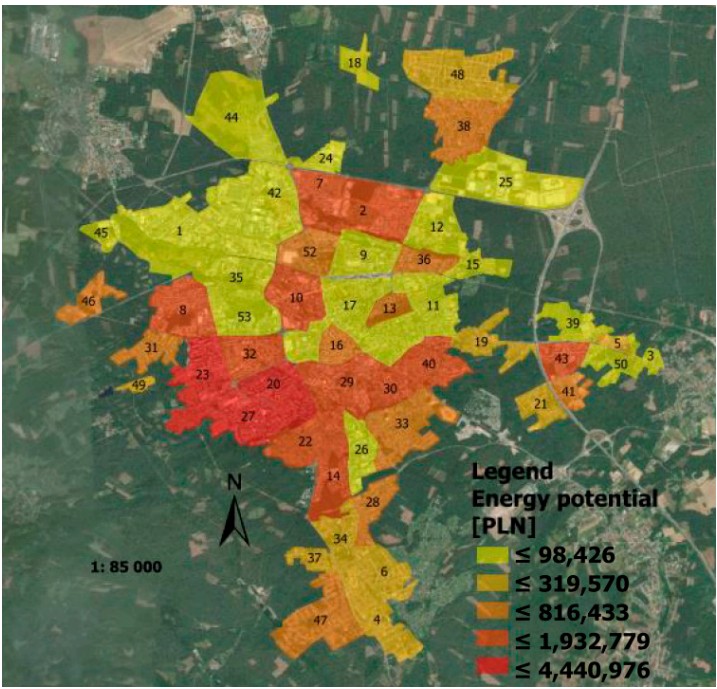

**Figure 8.** Energy potential for the Variant 2.

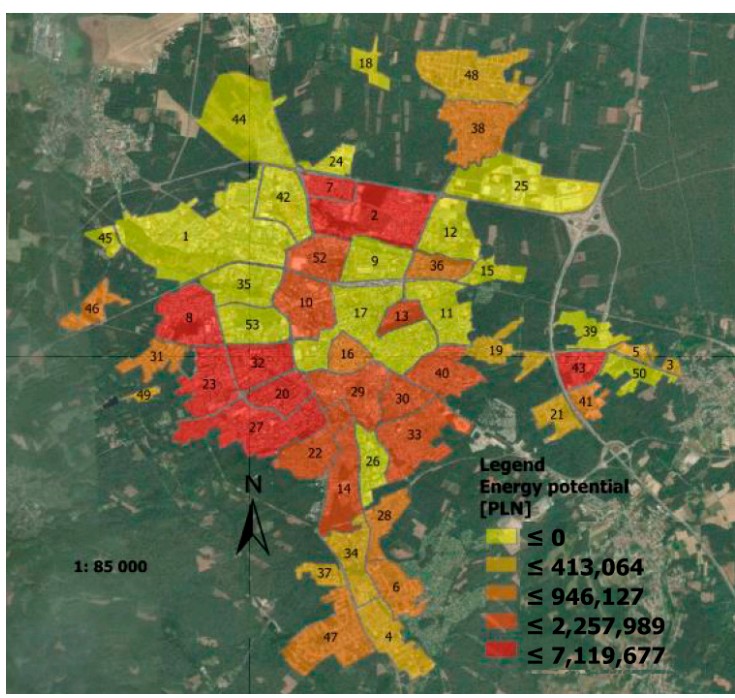

**Figure 9.** Energy potential for the Variant 3.

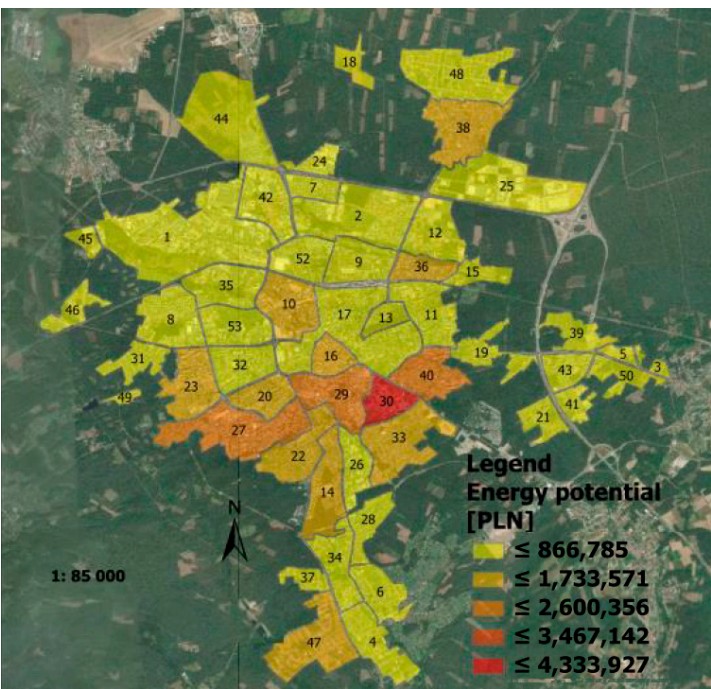

**Figure 10.** Energy potential for the Variant 4.

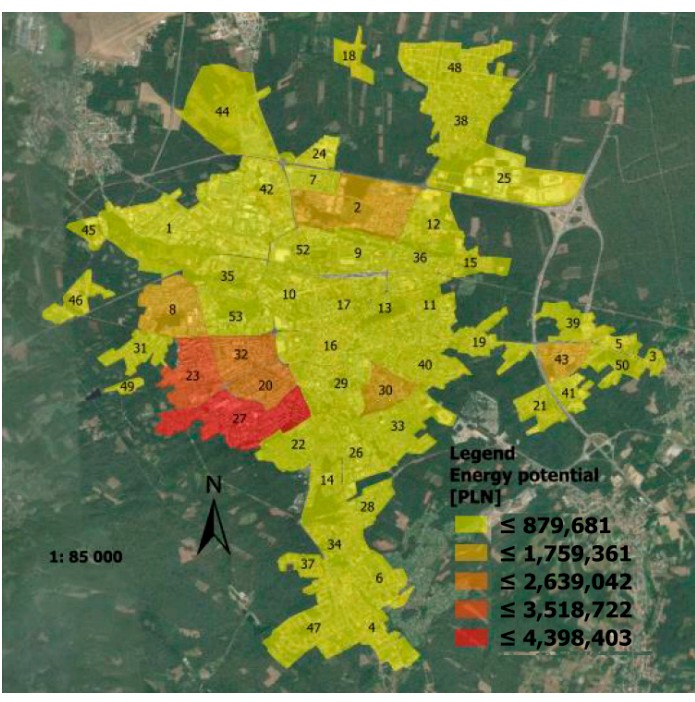

**Figure 11.** Energy potential for the Variant 5.

## 3.1. Discussion

Neves and team proposed a decision support methodology for supporting local sustainable energy planning processes. The research covered the entire energy planning process—from structure to a selection of an action plan, and the conclusions presented an assessment of alternative scenarios based on many strategic goals and preferences of local actors [46]. The choice of scenarios can avoid one-sector solutions and encourage many development trends, as described by Stanganelli and team [47]. By supporting the decision-making process with scenarios, you can face uncertainty about

future changes by creating preconditions for various possible trends in which successes will be chosen over time based on their ability to deal with economic, social, and environmental sustainability.

Interesting research on how to select and provide information on climate change with traditionally uninterested recipients was presented by Whitmarsh and Corner and was about testing several narratives for avoiding energy waste and advantages based on national low-carbon technologies. They also tried to find out why some concepts and topics are more effective, as well as how different elements of the narrative, e.g., the source of the message, determine its acceptance and how health and economic aspects can be omitted [48]. An interesting solution that helps reduce the impact of urban areas on climate change is also monitoring energy consumption in buildings through the use of automatic thermal modeling of building structures. This approach proposed in previous studies [49,50] creates new challenges for the community of incorporating existing GIS vector data and engages the public to provide voluntary information on urban facilities from which a new knowledge base is created to support urban energy efficiency.

Kim and his team, researching the possibility of reducing greenhouse gas emissions by increasing the energy efficiency of public buildings in the US, have shown that energy policy decisions require the involvement of many stakeholders as well as many related factors. As part of the analyzes carried out by researchers, decision-making factors were divided into five main categories: economic feasibility, environmental impact, institutional features, impact on users, and technical practicality. The use of event scenarios made it possible to present a method of approaching the complexity of the issue, cause and effect relationships, and the manner of executing tasks aimed at developing a management method for improving the energy efficiency of buildings [51].

### 3.2. Summary of Research

The research presented above confirms the thesis presented that the huge importance of achieving the goal of reducing energy consumption and stopping unfavorable climate change depends to a great extent on decision-makers. The created DGIS decision support and visualization system can help in developing a scenario of actions that meet the goal set. This document describes an innovative decision support methodology for local energy consumption planning that focuses on modeling energy-saving potential and assessing alternative scenarios based on many strategic goals and preferences of local decision-makers. According to the summary results presented in Table 4, there are visible differences in the costs of making decisions. In Option 5, hypothetical decisions taken by decision-makers preferring objectives mainly related to environmental protection are the least economically advantageous. This suggests the need to further link environmental goals with financial support for European programs at a local level. A broader view of environmental protection in the context of climate change will not be taken into account without the support of deliberate external measures.

On the other hand, the best decision-maker according to the financial criterion related to the energy-saving potential is the local government, which in summation, results in the best outcome. This suggests that local government decision-makers are the best trustees of budget revenues from residents' taxes. That result confirms that the adopted course of action and the goal of decisions taken at the local level are balanced and balanced towards maximizing financial effects. The summation result also indicates that the assumptions for equal weighting of all criteria presented in Option 1 do not lead to maximization of profits at all. The presented study proves once again that the strategies developed in the clash of many possible scenarios can bring multiplied profits of actions for specific purposes.

## 4. Conclusions

The article presents an innovative decision support system using multi-criteria analysis and GIS (DGIS), focusing on modeling energy-saving potential in urban areas, assessing alternative scenarios built based on many criteria, and preferences of decision-makers. Based on the multi-criteria assessment, choices were made regarding energy planning for decision-makers using different value schemes and with different local preferences. The innovation presented in the work concerned two

aspects: controlling the decision-making process (control and selection of information, and directing tools and resources to the eligible people) in a way that allows the highest possible energy savings and support with GIS technology; quick visualization of results allowing analysis of variants and tips for people decision-making. The content and form of information provided about the object should be adapted to the perception, decision-making and performance capabilities of the recipient. If the amount of information is too large, it may turn out that it is not fully used and it may bring losses arising during the implementation of the task. However, when the amount of information is too small, decisions are usually not optimal, which also increases losses.

The proposed approach allowed the estimation of energy-saving potential in the city, taking into account various criteria, including the possibility and willingness to invest in renewable energy by various groups of decision-makers. The scenario under analysis confirmed that one can try to predict the future by developing various likely but conflicting options and analyzing the costs of their implementation. Also, based on the scenarios, you can try to define the strategic goals of local energy policy, which is the basis for future research. The strategic goal should be to increase the use of renewable energy sources in urban areas. This is particularly important in developing countries where energy consumption from conventional sources is very high. At the same time, there are electricity shortages. Urban areas should have plans for the use of renewable energy sources in the event of a shortage [52]. And in this context the role of the decision-maker, in particular, the local government, becomes important. Besides, the results presented in the scenarios must be presented quickly and legibly to decision-makers.

The article presents the DGIS system supporting decision making by visualizing options for improving the energy efficiency of quarters in the city using GIS tools. The variants include several spatial energy mapping functions related to the choice of criteria depending on the decision-maker (filtering of input information from real studies on categories of buildings and other information to obtain a more understandable image that allows easier tracking of changes in energy consumption in the city). Further different key performance indicators are integrated to provide advanced information supporting decision making on energy efficiency measures. Based on the conducted research, it can be concluded that the combination of mathematical methods of multi-criteria analysis and GIS tools allows you to build an efficient system that can be a support in the decision-making process related to increasing the energy efficiency of urban areas as well as data analysis and construction of energy scenarios, including those taking into account the use of RES.

The approach proposed in the article is part of the new data processing strategies to build the most favorable energy scenarios in urban areas. It is also beneficial from energy planning related to the choices affecting the future of local societies, made by various entities whose preferences have been taken into account in the process of multi-criteria analysis. In future work, consideration should be given to creating a geoportal (open-source) that would implement the developed methodology and become a tool supporting decision-making processes for private or public entities.

Integrated management is the best form of development control. It is a complicated process, taking into account many factors and conditions, as well as the interests of those involved. Decision making should be based on a comprehensive analysis of conditions, setting goals, identifying potential and considering alternative solutions, and assessing the consequences of specific actions. In this context, decision making should be considered as a process that offers the opportunity to integrate actions between authorities and stakeholders. The model proposed in the work fits in with the above ideology regarding the method of managing the urban system.

Today's city is conditioned by an unlimited number of factors on which local authorities partly have no influence. According to this idea, the purpose of current planning is to safeguard the urban resilience system, undesirable interference, and shock. While local authorities have the opportunity to shape the appearance of the city, in the case of its mode of operation, their impact is highly limited. The idea of a self-organizing city is not the result of local government activity. The stability of urban structures is the result of interdependence and compiled structures functioning in local communities.

Taking this into account, cities should allow for continuous adjustment of plans, depending on changes in the operating conditions of cities. They should also combine spatial policy with economic and social policy, and thus use instruments that support seeing the whole process even when making current decisions regarding, e.g., energy efficiency of the buildings.

**Author Contributions:** Resources, M.S. (Małgorzata Sztubecka), M.S. (Marta Skiba), M.M. and A.B.-K. All authors have read and agreed to the published version of the manuscript.

**Funding:** This research was not financed from external sources.

**Conflicts of Interest:** The authors declare no conflict of interest.

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
