# Peer review of "An Innovative Decision Support System to Improve the Energy Efficiency of Buildings in Urban Areas"

_remotesensing, doi:10.3390/rs12020259_

Round 1

Reviewer 1 Report

The paper “Innovative decision support system to improve the energy efficiency of buildings in the urban área” presents an innovative decision support system using multi-criteria analysis and Geographic Information Systems (Decision Support System + Geographic Information Systems = DGIS) for planning urban development.

It is of great interest and may constitute an important methodological advance, however before being considered for this journal, several changes need to be made.

-A background section must be present in the paper. This is a methodological paper, so: What is the main contribution?

-The study area section is poor and need to be improve.

-Section 2.2. and 2.3 is only an introduction of GIS and Multi-criteria Analysis. This section does not explained the methodology. Although figures 2 and 3 are clearly presented, there is no explanation of the methodological phases. Furthermore, there is a confusion between results and methodology. I can see in the results, several methodological paragraphs.

-Figure 1. The scale is blurred. In general the image is blurred. Consider deleting the image entitled "Zielóna Gora". I think that zooming in 1 and 3 is sufficient

- Figures 7-11 need scale and North symbol. Legends needs a title. Also, consider to add a decimal comma or point.

-Figure 7. Why are there six decimal zeros in the legend?

- Which reference coordinate system was used?

In general, this reviewer congratulates the authors for the article, which is certainly interesting. However, its content needs to be improved and reordered.

The positive aspects should be highlighted and the methodology should be structured to improve understanding.

Author Response

We are grateful to the reviewers for their insightful and constructive comments on our manuscript entitled “Innovative decision support system to improve the energy efficiency of buildings in the urban area” (Manuscript ID: remotesensing-685712). We greatly appreciate the reviewers for their time and effort with constructive suggestions and comments. In the revised manuscript, we have attempted to address all the comments and issues raised by reviewers, which we hope to meet the reviewers' expectations. We have also addressed each of the points referred by them with sincerity and a detailed response addressing individual comment is presented below.

Reviewer 2 Report

This paper proposes the development of an innovative decision support system using multi-criteria analysis and Geographic Information Systems (DGIS) for planning urban development. The proposed decision support system provides information to energy consumers about the location of energy efficiency improvement potential. The proposed DGIS system can be used by local decision-makers, allowing better action to adapt cities to climate change and to protect the environment. This approach is part of new data processing strategies to build the most favorable energy scenarios in urban areas.

On the whole, the subject of the paper is presented in a comprehensive manner. However, I still have the following suggestions:

(1)There are some abbreviations and no full explanation. For instance, “CHP” , similar mistakes can be found throughout the paper, please check it. 

(2)   At line 146, there is a problem with the expression format of the matrix Z and it needs to be modified.

(3)   "Decision Support System + Geographical Information Systems = DGIS" has been repeated too many times. It is recommended that it only need to be introduced once in the main text, and the authors use abbreviations directly in the subsequent text, please modify it.

(4) In lines 138-145, the three parameters only are defined and the range of values or computational formula for the parameters is not explained, and please describe in detail.

(5) The algorithm in Figure 2 is not explained, and please explain it in detail.

Author Response

(The authors gave the same response as above.)

Reviewer 3 Report

The authors aim to present a decision support system using multi-criteria analysis, Decision Support System, and Geographical Information System, although the innovation that the title stresses (why?) is not that clear.

The problem domain is interesting, and the research gap is also clear: to provide tooling for decision makers in multi-criteria analysis in order to improve the energy efficiency of buildings in urban areas. The language of the article is proper.

The main problem with the article is the lack of the clear description related to the analysis method. What are the inputs, what are the calculation formulas (or other algorithms, methods) at each step? Without this it is hard to judge how this decision making system works and whether it provides meaningful results. Equations 1-5 are very generic, they are merely definitions, really - but no calculations, algorithms, methods, or pseudo-codes are presented that uses these defined items.

Tables II, III and IV are supposed to be the main results, but is is not described at all, how these numbers were calculated, and how can we judge these are correct even in their orders of magnitude. How should the indicators in Table II or III be interpreted? What do they mean? This article should be self-contained in the sense that these indicators (and their calculation, as well as validation) are explained.

The conclusion itself is far-fetched given that the results it references are not contained in the article. Each fact that the conclusion states should be either already clearly described in the article, or referenced from a different article. Acutally, the Related works towards the end of the article, as part of the discussion - should be at the beginning of the paper as Related Works, which then proves that there is a research gap that previous work have not fully covered, and this article aims for that.

A minor issue in the Introduction that references [10-17], [20-24], [25-27], [28-30] should be individually described. What are their contributions? Or these are just examples? (Why exactly these?) A few words on each would help the reader understanding the research gap, again. An even more minor issue is that "Currently (as of December 2018)" is an outdated statement in this form.

Author Response

(The authors gave the same response as above.)

Round 2

Reviewer 1 Report

The paper “Innovative decision support system to improve the energy efficiency of buildings in the urban área” presents an innovative decision support system using multi-criteria analysis and Geographic Information Systems (Decision Support System + Geographic Information Systems = DGIS) for planning urban development.

The authors have made a great improvement to the article. The introduction has been greatly improved, and a background section has been added.

The study area has also been improved, although the map still looks "fuzzy". I think it should not be difficult for the authors to export the map with higher resolution.

Also, the methodology has been improved, although here I denote the need for it to be described more clearly. This point is crucial. Please, check english language in Methodology section, there are some typos in the text.

Minor corrections

-Consider changing " STep 2. Definition" x Step 2. "Defining" Line 225

-Regarding the maps must, please use only one type of scale (graphical or numerical), and repace Kilometry for km. You can also improve the scale by reducing the number of “sections in the graphical scale” and changing decimal comma by “points”.

- Regarding the reference coordinate system, this sentence would be sufficient:

The work uses the PL-ETRF89 geodetic reference systems, which are the mathematical and physical implementation of the European earth reference system ETRS89. This reference system and coordinate system used are legally defined in Poland in the 342 Regulation of the Council of Ministers [43].

Author Response

We are grateful to the reviewers for their insightful and constructive comments on our manuscript entitled “Innovative decision support system to improve the energy efficiency of buildings in the urban area” (Manuscript ID: remotesensing-685712). We greatly appreciate the reviewers for their time and effort with constructive suggestions and comments. In the revised manuscript, we have attempted to address all the comments and issues raised by reviewers, which we hope to meet the reviewers' expectations.

Reviewer 3 Report

The paper has improved significantly since the last version. It can be accepted after a few very minor corrections.

The discussion after results (with the different related works) can remain at the end of the document (as a discussion), but it somewhat comes suddenly after the results (own diagrams and their description), that "Neves and the team proposed a decision support methodology...". This still feels strange that these other researches are coming so suddenly after own results. The authors should consider making a subsection here, or even a section (no pressure, but worth considering).

Lines 11-112: With the correction applied, would this appear as GIS (DGIS). ? It is strange this way... similarly to lines 504-505

Figure 2: text is underlined with red; the figure should be cut and pasted without this language check marker...

In the Conclusion this statement is striking: "The scenario analysis confirmed that one can try to predict the future by" ... maybe >>try to predict future energy usage << (or energy saving, or...); but not the future as a whole. (Sorry,... I know.)

Author Response

(The authors gave the same response as above.)
